# BLBP Is Both a Marker for Poor Prognosis and a Potential Therapeutic Target in Paediatric Ependymoma

**DOI:** 10.3390/cancers13092100

**Published:** 2021-04-27

**Authors:** Durgagauri H. Sabnis, Jo-Fen Liu, Lucy Simmonds, Sophie Blackburn, Richard G. Grundy, Ian D. Kerr, Beth Coyle

**Affiliations:** 1Children’s Brain Tumour Research Centre, School of Medicine, University of Nottingham Biodiscovery Institute, Nottingham NG7 2RD, UK; dgs3988@gmail.com (D.H.S.); Jo-fen.Liu@nottingham.ac.uk (J.-F.L.); lucysimmonds@nhs.net (L.S.); sophie.blackburn1@nhs.net (S.B.); richard.grundy@nottingham.ac.uk (R.G.G.); 2School of Life Sciences, QMC, University of Nottingham, Nottingham NG7 2UH, UK; ian.kerr@nottingham.ac.uk

**Keywords:** BLBP, ependymoma, brain tumor, PPAR, docosahexanoic acid, invasion, drug resistance, FABP7

## Abstract

**Simple Summary:**

Ependymomas are the second most common paediatric brain tumour, and the 5-year survival rate remains no higher than 50%. Identifying new prognostic markers and targets for therapy in ependymoma is an important area of research. In this study, we demonstrate that brain lipid binding protein (BLBP, FABP7) is expressed in a sub-population of cells in ependymoma patient samples, consistent with it being a cancer stem cell marker. BLBP expression was associated with both significantly reduced event free survival and overall survival. BLBP can be functionally inhibited by PPAR antagonists, and we demonstrated that these antagonists can reduce ependymoma cell migration, invasion and chemo-resistance. The BLBP ligand docosohexanoic acid also attenuated these three hallmark characteristics of ependymomas, leading us to conclude that BLBP is not just a prognostic marker for poor ependymoma survival, but that it represents a druggable target in ependymoma therapy.

**Abstract:**

Paediatric ependymomas are aggressive, treatment-resistant tumours with a tendency towards relapse, consistent with a sub-population of therapy-resistant cancer stem cells. These cells are believed to derive from brain lipid binding protein (BLBP)-expressing radial glia, hence we proposed that BLBP may be a marker for ependymoma therapy resistance. BLBP protein expression correlated with reduced overall survival (OS) in patients from two trials (CNS9204, a chemotherapy-led infant trial—5 y OS 45% vs. 80%, *p* = 0.011—and CNS9904, a radiotherapy-led trial—OS 38% vs. 85%, *p* = 0.002). All ependymoma cell lines examined by qRT-PCR expressed *BLBP*, with expression elevated in stem cell-enriched neurospheres. Modulation of BLBP function in 2D and 3D assays, using either peroxisome proliferator activated receptor (PPAR) antagonists or BLBP’s fatty acid substrate docosahexaneoic acid (DHA), potentiated chemotherapy response and reduced cell migration and invasion in ependymoma cell lines. BLBP is therefore an independent predictor of poor survival in paediatric ependymoma, and treatment with PPAR antagonists or DHA may represent effective novel therapies, preventing chemotherapy resistance and invasion in paediatric ependymoma patients.

## 1. Introduction

Ependymomas are the second most common malignant paediatric brain tumour. Almost 90% of paediatric ependymomas occur intracranially, with two thirds of them arising infratentorially from the posterior fossa (PF), and the remaining one third from the supratentorial (ST) compartment [1]. Forty percent of cases remain incurable and 5-year survival in infants with ependymomas stands at a dismal 40–52% [2,3]. Thus far, only complete surgical resection has been regarded as a positive marker of clinical outcome in ependymomas [4], with genetic studies yielding just three other progostic markers, namely ABCB1 [5], gain of chromosome 1 q [6,7] and the RELA fusion, which specifically occurs in aggressive supratentorial ependymomas [8,9].

Chemo-resistance and local invasion are hallmarks of ependymomas which contribute to their recurrence in 50% of patients [10,11]. Both these features have been correlated with the existence of a sub-population of cancer stem cells (CSCs) in other tumour types [12]. We have previously demonstrated the presence of a sub-population in ependymomas and have also confirmed its functional role in ependymoma chemo-resistance [13]. Markers of this sub-population which can act as either prognostic indicators or targets for therapy would be valuable in the management of paediatric ependymoma.

Taylor et al. identified brain lipid binding protein (BLBP; also known as fatty acid binding protein 7) expressing transformed radial glia as the stem cell of origin in ependymoma [14]. BLBP is an intracellular chaperone for fatty acids (FA) and also plays a key role in their storage and metabolism [15,16]. Its primary binding substrates include the polyunsaturated fatty acids docosahexaneoic (DHA; omega-3) and oleic acid (omega-9) [17]. DHA is essential for foetal brain development and BLBP is a developmental marker expressed by the radial glia during neurogenesis [15]. BLBP expression has been linked to poor clinical outcome in several aggressive cancers including glioblastoma multiforme (GBM) [18,19,20].

Although several knockdown studies have reported the potential functional role of BLBP in proliferation and invasion of cancer [18,21,22], the underlying mechanism remains unclear. Indeed, the identity of the FAs chaperoned by BLBP has been shown to modulate expression of several genes including BLBP itself, via interactions with different peroxisome proliferator activated receptor (PPAR) nuclear transcription factors [23] with different downstream effects. Thus, Martin et al. reported that whilst the omega-3 FA DHA modulated the transcriptional activity of PPARs to inhibit migration and proliferation in gliomas, the omega-6 FA arachidonic acid (AA) promoted these cancerous mechanisms [21].

In this study, we explore the prognostic as well as the functional role of BLBP in paediatric ependymomas. In addition to determining a correlation between BLBP expression and poor outcome in two independent paediatric ependymoma trial cohorts, we also found that inhibiting BLBP via PPAR antagonism or by the omega-3 FA DHA, targeted chemo-resistance, migration and invasion in paediatric ependymoma cell lines.

## 2. Results

### 2.1. BLBP Expression in a Sub-Population of Cells Correlates with Poor Prognosis

In order to assess whether BLBP was a marker for poor prognosis in ependymoma, we looked at expression across two trial cohorts. In the chemotherapy-led CNS9204 infant (under 3 years of age) trial, 47% (25/53) of patients were found to be BLBP-positive, and in agreement with the CSC hypothesis, only a sub-population of cells per positive sample showed cytoplasmic and nuclear BLBP staining (Figure 1a,b). In this trial, BLBP-positive patients had a significantly lower event free survival (5-year EFS: 27% versus 58%, *p* = 0.028) and overall survival (5-year OS: 44% versus 82%, *p* = 0.01) as depicted in Figure 1c,d. Furthermore, the feasibility of BLBP as an independent prognostic marker was assessed by employing the Cox-regression multivariate model. Surgical resection, tumour location and WHO grade were included as confounding factors in this model. As shown in Table 1, BLBP-positive patients were more susceptible to having an event (relapse or death) (hazard ratio 2.08; *p* = 0.04) and were more likely to die (Hazard ratio 2.81; *p* = 0.01) in comparison to those who were BLBP-negative.

In the radiotherapy-led CNS9904 trial consisting of patients between 3 and 21 years of age, 32% (12/38) of patients were BLBP-positive. There was no correlation between BLBP expression and reduced EFS (5-year EFS: 33% versus 61%, *p* = 0.106; Figure 1e); however, it was significantly associated with reduced OS (5-year OS: 38% versus 84%, *p* = 0.002; Figure 1f) in patients. After carrying out multivariate analyses employing the aforementioned confounding factors, BLBP was found to be a robust marker of reduced OS in the CNS9904 trial cohort (Hazard ratio 5.37; *p* = 0.004; Table 1). We also looked at BLBP with regard to resection status across both trials and showed that it was a significant prognostic marker irrespective of the extent of resection (see Appendix A). Finally, analysis of gene expression datasets for paediatric ependymoma molecular subgroups revealed high *BLBP* expression, with only the ST-EPN-YAP1 subgroup showing significantly lower expression, and that expression is increased in relapsed relative to primary samples from these subgroups (see Appendix A).

### 2.2. BLBP Is a Stem Cell Marker in Ependymoma

To identify an appropriate model for the investigation of BLBP function, we next characterised ependymoma cell lines. BLBP gene expression was assessed in five paediatric ependymoma cell lines using QRT PCR analysis (Figure 2). In comparison to the housekeeping gene *GAPDH*, varying levels of BLBP were recorded in all ependymoma cell lines.

The EPN1 and BXD-1425EPN cell lines expressed low but detectable levels of BLBP (Figure 2b). EPN1 was established in-house from a second recurrence, and analysis of samples taken from the primary tumour and two recurrences all showed low numbers of BLBP positive cells consistent again with the CSC hypothesis (Figure 2a). This expression pattern was also maintained in an orthotopic xenograft of the tumour (Figure 2a). Intermediate levels of BLBP were expressed in the EPN7 and EPN7R cell lines, which were derived from the primary and recurrent ependymomas of the same patient, respectively. The highest expression of BLBP was recorded in the DKFZ-EP1 cell line. As expected for a stem cell marker, when the DKFZ-EP1 cell line was grown as neurospheres (DKFZ-EP1NS), there was a concomitant five-fold increase in BLBP expression (*p* < 0.001; Figure 2c).

### 2.3. BLBP Inhibition by PPAR-Antagonists or by Natural Ligands Reduces Stem Cell Viability and Invasion in Ependymoma Cell Lines

To elucidate the functional role of BLBP in paediatric ependymomas we compared high and low expressing ependymoma cell lines (DKFZ-EP1 and the BXD-1425EPN, respectively). We treated both cell lines with three PPAR antagonists; the dual PPAR-(γ/δ) antagonist FH535, the PPAR-δ antagonist GSK0660 and the PPAR-γ antagonist GW9662, to assess their effect on BLBP expression. As depicted in Figure 3, each of the three PPAR antagonists knocked down BLBP expression in the BLBP^lo^ BXD-1425EPN cell line to barely detectable levels (Figure 3a(i)). In the BLBP^hi^ DKFZ-EP1 cell line, the PPAR-γ antagonist GW9662 resulted in almost complete loss of BLBP expression, the PPAR-δ antagonist GSK0660 resulted in a partial but significant loss, and the dual PPAR-(γ/δ) antagonist FH535 resulted in non-significant effects on BLBP gene expression (Figure 3a(ii)). Longer incubations resulted in a persistent reduction in BLBP expression (see Appendix A online). We then investigated whether these effects on BLBP gene expression would result in alterations to viability. In stem cell relevant clonogenic and neurosphere assays carried out on BXD-1425EPN and DKFZ-EP1 cell lines, respectively, both dual PPAR-(γ/δ) antagonist FH535 and PPAR-γ antagonist GW9662 significantly reduced cell viability (Figure 3b,c).

BLBP expression has been correlated with the migratory nature of glioma cell lines [24]. In a 2D wound healing assay, each of the three PPAR antagonists significantly inhibited migration in both cell lines in comparison to the vehicle treated control, resulting in increased t_1/2_ for wound closure (Figure 4a(i,ii)). Treatment of in vitro models of cancer with DHA, an omega-3 fatty acid that binds to BLBP with high affinity [15], has been shown to inhibit proliferation and migration [24,25,26]. Similarly, we showed here that treatment with DHA significantly increased the t_1/2_ for wound closure in both ependymoma cell lines, as depicted in Figure 4b. In a 3D spheroid invasion assay, GW9662 significantly inhibited invasion of BXD-1425EPN cells through extracellular matrix, whereas the dual PPAR antagonist FH535 (despite only a modest effect on *BLBP* expression) resulted in almost total inhibition of invasion (Figure 4c(i,ii)). Similarly, DHA also significantly inhibited the ability of the BXD-1425EPN cell line to invade through the BME (Figure 4c).

### 2.4. BLBP Inhibition Targets Chemo-Resistance in Ependymoma Cell Lines

Both local invasion and chemotherapy resistance are hallmarks of poor outcome in paediatric ependymoma. We therefore also ascertained whether the inhibition of BLBP expression and/or function made the cell lines more susceptible to chemotherapy. In these experiments, we employed the PPAR-γ antagonist GW9662, which had a consistent inhibitory effect on BLBP expression and survival in both cell lines. In a stem cell-relevant clonogenic assay, pre-treatment of the BLBPlo BXD-1425EPN cell line with GW9662 for 48 h followed by treatment with etoposide and vincristine at their IC50 values [5] potentiated response to both drugs (Figure 5a(i,ii)).

Moreover, in our study, DHA also enhanced the cytotoxic effect of etoposide and vincristine in the BXD-1425EPN cell line in a clonogenic assay (*p* < 0.001, Figure 5b(i,ii) [15]), in accordance with previous work on BLBP in medulloblastoma and glioma [24,25,26,27,28,29]. Indeed, combining either DHA or GW9662 with etoposide resulted in complete loss of clonogenic potential in this cell line.

## 3. Discussion

BLBP expression has been correlated with poor clinical outcome in several aggressive forms of cancer, such as renal cancer [22], hepatocellular carcinoma [30] and aggressive triple-negative breast cancer [31]. Since CSCs contribute to the aggressive nature of tumours and BLBP was a marker for these CSCs in ependymoma, we proposed that BLBP expression could be of robust prognostic value in ependymoma, a tumour for which prognostic markers are urgently required. This hypothesis was supported by our results which show that BLBP protein expression in a sub-population of patient samples was indeed an independent prognostic marker for EFS and OS in the chemotherapy-led CNS9204 trial and for OS in the radiotherapy-led CNS9904 trial. Importantly, we were also able to demonstrate that there was no effect modification by resection status. Methylation subgroups had been assigned for 48 of the patients included in this study, so we were also able to determine if BLBP expression defined a specific subgroup. BLBP was expressed in three subgroups (EPN PFA, ST-EPN-RELA and EPN YAP); although percentages were only meaningful in EPN PFA (50% positive; 13/26) and ST-EPN-RELA (43% positive; 3/8). Whilst the low patient numbers do not currently enable us to perform sub-group specific confirmation of BLBP’s diagnostic potential, it is clear that we have identified a novel prognostic marker for this tumour.

BLBP is a marker for radial glia (RG), which are precursors for several neuronal cells, including ependymal cells [15]. Taylor et al. demonstrated that BLBP expressing transformed radial glia were the CSCs in ependymomas, since they expressed cell surface markers such as CD133, formed self-renewing neurospheres in culture, and were able to recapitulate murine orthotopic tumours [14]. Our study supports the argument that BLBP is indeed a CSC marker in ependymoma. Firstly, we found that BLBP protein was expressed in a sub-population of cells in patient samples. Secondly, we observed that this expression was maintained at recurrence and in orthotopic tumours and, thirdly, that BLBP was expressed in a panel of ependymoma cell lines with expression increased in stem cell-enriched neurospheres. All of these findings led us to propose that if maintenance of BLBP expression was important, then manipulation of its function may be of therapeutic value.

Fatty acid-loaded BLBP modulates gene expression through interaction with PPARs; indeed, BLBP expression itself is regulated in this way [32]. We postulated that chemical inhibition of BLBP may provide a more effective route to be able to translate our findings to the clinic. We therefore targeted BLBP in two ways; by application of its direct fatty acid ligand DHA or by application of PPAR antagonists. In agreement with the results of De Rosa et al., incubation of ependymoma cell lines with subtype specific or dual-specificity PPAR antagonists resulted in reductions in BLBP expression [32]. Reports regarding the effect of BLBP inhibition on proliferation in in vitro models of cancer have been controversial [22,32,33]. In our study, the dual PPAR-(γ/δ) antagonist FH535 and the PPAR-γ antagonist GW9662 inhibited viability in stem cell-relevant assays. This potent effect on cell viability might be a result of the enriched BLBP expression in ependymoma CSCs.

Ependymoma tumours are difficult to treat chemotherapeutically and show a tendency to invade locally. We assessed whether there may be a role for BLBP in either or both of these processes. GW9662 significantly potentiated the response to both drugs in the BLBP^lo^ BXD-1425EPN cell line in a stem cell-relevant clonogenic assay, indicating that treatment with this PPAR-γ antagonist, GW9662, would be a useful strategy in targeting this subpopulation of treatment refractory transformed RGs. In agreement with several previous studies [21,22,23,32], we found that BLBP inhibition through PPAR antagonists reduced the migratory and invasive capacity of ependymoma cells.

As described in the introduction, and demonstrated here, the nature of BLBP’s ligand and BLBP:PPAR interactions can determine migration and proliferation phenotypes (Appendix A). We therefore believe that the best approach to elucidating BLBP’s therapeutic role will be to determine how to direct its functional regulation towards pathways that inhibit tumour growth and migration, rather than to block BLBP expression completely.

DHA is a popular dietary supplement with several health benefits, including maintenance of normal brain function [34]. There have been several studies which highlighted the efficacy of DHA in targeting chemo-resistance [27,28,29], proliferation and invasion in in vitro as well as in in vivo models of several cancers [25,26]. In the BLBP^lo^ BXD-1425EPN cell line, DHA co-treatment enhanced the response to both etoposide and vincristine in a stem cell relevant clonogenic assay. DHA reduced the migratory capacity of cells of both cell lines and marginally but significantly reduced the ability of the BLBP^lo^ BXD-1425EPN cells to invade through BME.

This anti-migratory effect of DHA has been proposed to be due to the inhibition of pro-migratory genes such as COX-2, CD44 and adhesion molecules ICAM^30^; however, irrespective of what the downstream mediators of this process are, these findings suggest that fatty acids may mediate many key processes in ependymoma pathogenesis. Importantly, this may support the recent benefits associated to a ketogenic diet; however, the application of PPAR antagonists or DHA may prove more palatable. In support of this approach, a phase II clinical trial has recently been published proposing DHA supplementation alongside chemotherapy in breast cancer [35]. A similar approach could easily be applied in patients with brain tumours.

## 4. Materials and Methods

### 4.1. Trial Cohorts and Immunohistochemistry (IHC)

The tissue microarrays (TMAs) screened in this study were obtained from the CCLG/SIOP Infant Ependymoma (CNS9204) and SIOP Ependymoma I clinical trial (CNS9904). Patients in the CNS9204 trial cohort were up to 3 years of age at diagnosis. They received chemotherapy for approximately one year after diagnosis and radiotherapy was only given at relapse. The CNS9904 cohort consisted of patients who were only given chemotherapy if incomplete resection was determined after a second attempt at surgery. Clinicopathological details of both these trials have been previously described [5,36]. Sections from the mouse orthotopic xenograft model were produced in a previous study [13].

Immunohistochemistry staining of TMAs was performed using rabbit anti-BLBP polyclonal antibody (ABN14, Millipore, Darmstadt, Germany; 1:500) with the Dako Chemate Envision Antigen Detection kit (Dako, Thetford, UK) as previously described [37]. Only a sub-population of cells per positive sample showed cytoplasmic and nuclear BLBP staining; hence, samples were scored as a binary variable (positive or negative). The Kaplan–Meier method was used to examine the association between the BLBP expression status and overall survival (OS; time between the date of diagnosis and death) as well as event free survival (EFS; time between date of diagnosis and first event (recurrence/death)). Patients still alive at the end of the study were censored at the date of the last follow-up. The differences in survival between expression groups were estimated using the long-rank (Mantel–Cox) test. The Cox proportional hazard regression model was used to measure the effect of multiple confounding factors in order to test if BLBP had an independent correlation with poor survival. IBM SPSS 22.0 for Windows (IBM Corp. Armonk, NY, USA) was used to carry out the survival analyses.

### 4.2. Ethics Declarations

These studies, and the experimental protocols required, were reviewed and approved by the National Research Ethics Service Committee East Midlands—Nottingham 2, and were therefore performed in accordance with the ethical standards laid down in an appropriate version of the 1975 Declaration of Helsinki, as revised in 1983.

### 4.3. Consent to Participate

For all patients, informed consent was obtained from the patient, or a parent and/or legal guardian where the patient was under 18 years of age, prior to their inclusion in the study.

### 4.4. Cell Lines

This study is based on experiments performed on 5 paediatric ependymoma cell lines. Out of these, the EPN1R, EPN7 and EPN7R were established in-house and PCR single-locus-technology (Eurofins, Brussels, Belgium) confirmed that both lines match their tumour of origin. The BXD-1425EPN and DKFZ-EP1NS were previously derived and characterised by Dr Xiao-Nan Li, Baylor College of Medicine [38] and Dr Till Milde, DKFZ Heidelberg [39] respectively. All cell lines were grown as adherent monolayers in ‘tumour media’ as previously described [5]. The DKFZ-EP1 neurospheres (DKFZ-EP1NS) were cultured in ‘stem cell media’ as described [39]. Functional analyses were carried out according to the doubling time; in the case of the DKFZ-EP1 cell line, measurements were made at 48, 96 and 144 h, whilst in the BXD-1425EPN cell line, it was measured at 24, 48 and 72 h (each 1, 2 and 3 doubling times). DKFZ-EP1, EPN1 and BXD-1425EPN are supratentorial (ST) tumours with a C11orf95–RELA fusion, and so can be classified as ST-EPN-RELA. EPN7 (EPN7R is the recurrence) was derived from an anaplastic supratentorial tumour and does not harbour the fusion protein.

### 4.5. Real Time PCR

Real-time PCR analysis was performed as previously described [37]. *BLBP* primer sequences were forward 5′ GTTGTTAGCCTGGATGGAGAC 3′ and reverse 5′ CATAGTGGCGAACAGCAAC 3′. The relative *BLBP* mRNA expression level in comparison to *GAPDH* was calculated using the ΔCt method [40]. *GAPDH* was used as the housekeeping gene since the transcript levels were found to be stable across multiple passages in both adherent and neurosphere cultures.

### 4.6. Viability Assays

In order to assess the cytotoxic toxic effects of agents, clonogenic or stem cell-enriched spheroid assays were performed. All the reagents used in this study were bought from Sigma (St. Louis, MO, USA). Chemotherapy drugs were used at previously established IC_50_ concentrations (etoposide 19.2 µM and vincristine 23.5 nM) [5], dual PPAR-(γ/δ) antagonist FH535, the PPAR-δ antagonist GSK0660, and the PPAR-γ antagonist GW9662 were all used at 15 µM and DHA was used at 20 µM.

In the clonogenic assay, 600 cells/well of the BXD-1425EPN cell line were plated in a 6-well plate and the experiment was performed as described previously [37]. In each experiment, data were obtained from averaging the number of blue colonies (of over 50 cells) in duplicate wells of a 6 well plate over at least 3 independent experiments. In the stem cell-enriched spheroid assay, neurospheres were dissociated into single cells and 5000 cells were then plated in each well of an ultra-low attachment 96-well round bottom plate (Corning, Corning, NY, USA) in stem cell medium enriched with 20 ng/mL basic-fibroblast growth factor (bFGF, GIBCO) and 20 ng/mL epidermal growth factor (EGF, GIBCO). The cells were then centrifuged at 100 g for 5 min and incubated at 37 °C and 5% CO_2_. Once the stem cell-enriched spheroids had formed on day 3, they were treated with PPAR antagonists. Viability was assessed using Alamar Blue^®^ (Life Technologies, Carlsbad, CA, USA) and the fluorescence was measured on a FLUOstar plate reader (BMG labtech instruments).

### 4.7. Wound Healing Assay

A wound healing assay was performed to measure the migratory capacity of cells. After wounding, cells were treated either with vehicle (DMSO/H_2_O), PPAR antagonists or DHA, imaged and would closure quantified as previously described [5].

### 4.8. Three-Dimensional Spheroid Invasion Assay

The ability of the cells to invade was assessed by carrying out a 3D spheroid assay in ultra-low attachment 96-well round bottom plates using Cultrex^TM^. The experiments and analysis were performed as previously described [5]. The relative spheroid outgrowth (R) was calculated by taking the ratio of the area of the invasive edge to the area the spheroid.

### 4.9. Data Analysis

All data shown result from at least three independent experiments and unless otherwise stated, error bars represent standard error of the mean and have been calculated using Graphpad Prism Version 7.0 (GraphPad Software, La Jolla CA USA). Comparisons between data sets were performed using student unpaired *t*-test or ANOVA analysis. *p*-values are presented as * < 0.05, ** < 0.01, *** < 0.005, **** < 0.001.

## 5. Conclusions

In conclusion, in this study, we were able to provide the first evidence of the prognostic value of BLBP in ependymoma. Furthermore, we also propose that both the PPAR-γ antagonist, GW9662, and omega-3 fatty acid, DHA, would counter the chemo-resistant and invasive behaviour of ependymoma.

## Figures and Tables

**Figure 1 cancers-13-02100-f001:**
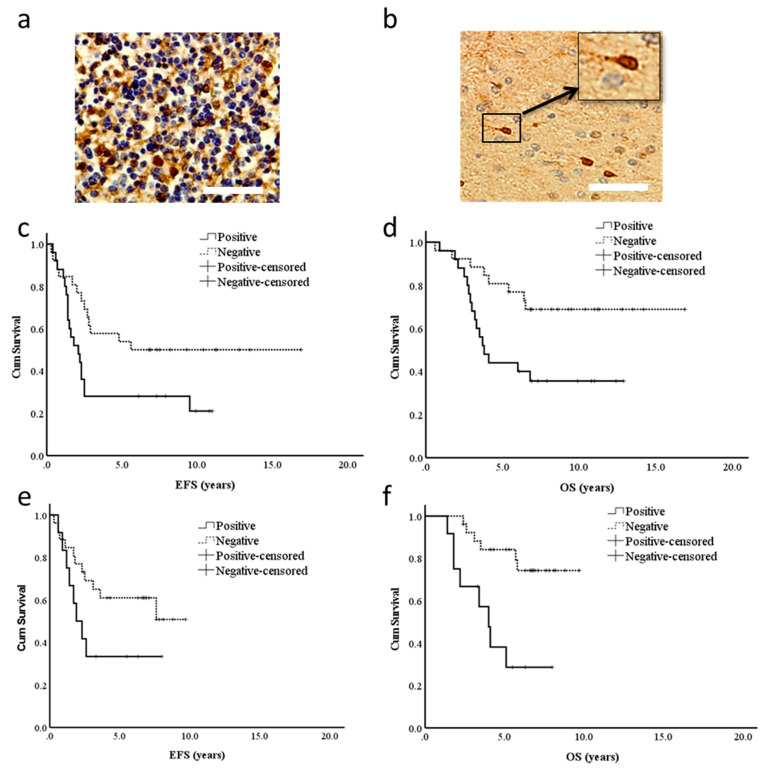
BLBP expression in a sub-population of cells is associated with poor prognosis in ependymoma patients. (**a**) In normal foetal brain tissue, 70–80% of cells expressed BLBP. (**b**) A BLBP-positive paediatric ependymoma demonstrating typical staining of a sub-population of cells. Scale bars represent 50µm. (**c**,**d**) In the chemotherapy-led CNS9204 trial, BLBP-positive patients had a significantly reduced event-free survival (**c**) (5-year EFS 27% versus 58%, *p* = 0.028) and overall survival (**d**) (5-year OS 44% versus 82%, *p* = 0.01), respectively. (**e**,**f**) In the radiotherapy-led CNS9904, BLBP expression did not correlate with reduced event-free survival (**e**) (5-year EFS 33% versus 61%, *p* = 0.106); however, it was significantly associated with reduced overall survival (**f**) (5-year OS 38% versus 84%, *p* = 0.002).

**Figure 2 cancers-13-02100-f002:**
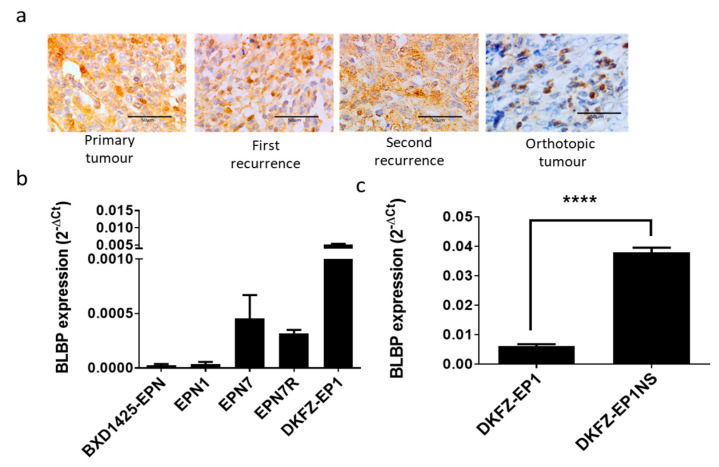
BLBP expression in cell lines and corresponding tumours. (**a**) BLBP protein was consistently expressed in a sub-population of cells in the primary and recurrent patient tumours as well as the EPN1orthotopic mouse tumour (derived from the second recurrence). Scale bars represent 50 µm. (**b**) Varying levels of *BLBP* gene expression, relative to *GAPDH*, were observed in each of the 5 paediatric ependymoma cell lines. (**c**) An approximate 5-fold increase in *BLBP* expression was observed when the DKFZ-EP1 cell line was grown as neurospheres (DKFZ-EP1NS) (**** *p* < 0.001; *n* = 3).

**Figure 3 cancers-13-02100-f003:**
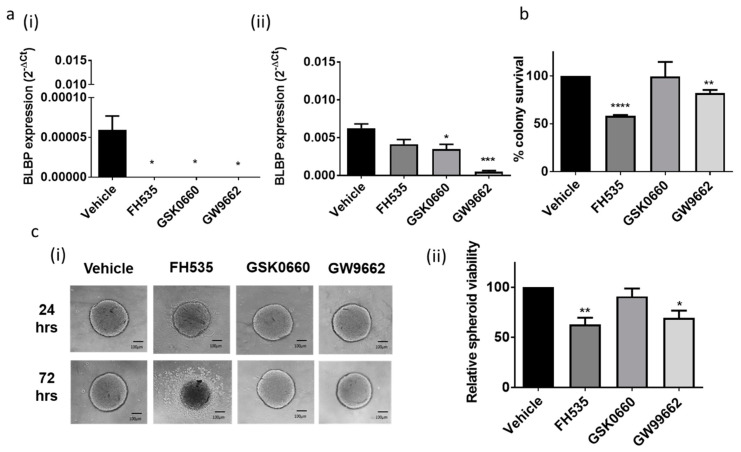
BLBP inhibition by PPAR antagonists inhibited viability in stem cell relevant assays. (**a**) Both the ependymoma cell lines were treated with the 3 PPAR antagonists. (**i**) In the BLBPlo BXD-1425EPN cell line, *BLBP* gene expression levels, relative to *GAPDH*, significantly diminished as a result of treatment with each of the 3 PPAR antagonists, whilst in the (**ii**) BLBP^hi^ DKFZ-EP1 cell line, the PPAR-γ antagonist GW9662 and PPAR –δ antagonist GSK0660 significantly inhibited *BLBP* expression. These results were recorded 24 or 48 h after treatment (one doubling time). (**b**) In BXD-1425EPN, the dual PPAR antagonist FH535 and GW9662 significantly reduced clonogenic survival by 42.5% and 18%, respectively, relative to vehicle controls. (**c**). In DKFZ-EP1NS, FH535 and GW9662 decreased stem cell enriched spheroid viability, depicted by a halo of dead cells around the spheroids (**i**), by 37% and 30%, respectively, compared to vehicle (**ii**). Scale bars represent 100 µm; *n* = 3. *p*-values are presented as * < 0.05, ** < 0.01, *** < 0.005, **** < 0.001.

**Figure 4 cancers-13-02100-f004:**
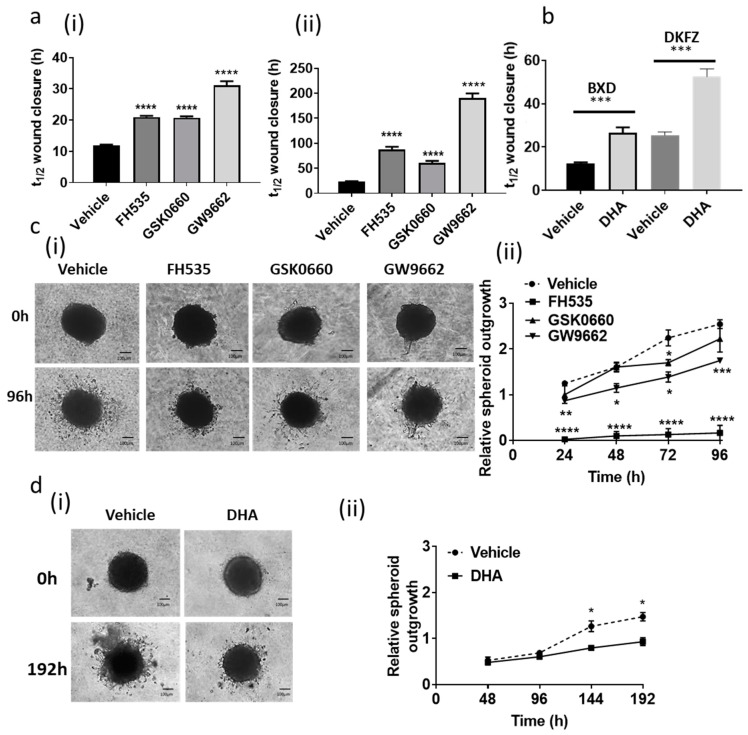
BLBP inhibition by PPAR antagonists and DHA decreased migration and invasion. (**a**) When treated with PPAR antagonists (15 µM), there was a significant reduction in the wound closure rate for both (**i**) BXD-1425EPN and (**ii**) DKFZ-EP1. (**b**) DHA treatment also significantly decreased wound closure rates in BXD-1425EPN (BXD) and DKFZ-EP1 (DKFZ) cell lines. (**c**) (**i**) Invasion of BXD-1425EPN spheroids through Cultrex^TM^ Basement Membrane Extract was distinctly reduced by the dual PPAR-(γ/δ) antagonist FH535 and the PPAR-γ antagonist GW9662. (**ii**) Quantification of the relative spheroid outgrowth confirmed a time-dependent significant reduction in invasion (*n* = 3). (**d**) (**i**,**ii**) BXD-1425EPN spheroid invasion was also reduced over 192 h following treatment with DHA. Scale bars represent 100 µm; *n* = 3. *p*-values are presented as * < 0.05, ** < 0.01, *** < 0.005, **** < 0.001.

**Figure 5 cancers-13-02100-f005:**
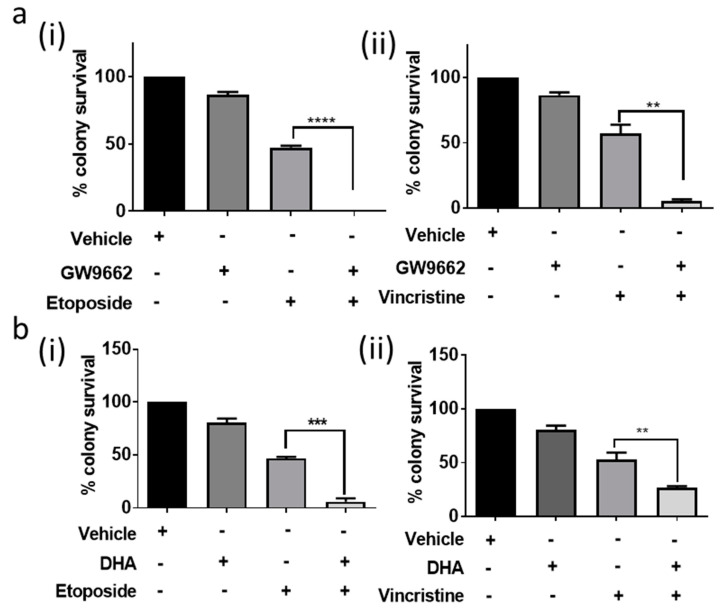
BLBP inhibition by PPAR antagonists and DHA decreased chemoresistance. (**a**) The PPAR-γ antagonist GW9662 potentiated response of BXD-1425EPN to (**i**) etoposide (*p* < 0.001) and (**ii**) vincristine (*p* = 0.002), at their respective IC50 concentrations in a clonogenic assay. (**b**) DHA co-treatment with etoposide (**i**) or vincristine (**ii**) decreased clonogenic survival of BXD-1425EPN to a greater extent than either individual treatment. *p*-values are presented as ** < 0.01, *** < 0.005, **** < 0.001.

**Table 1 cancers-13-02100-t001:** Multivariate survival analyses of BLBP expression in two ependymoma trials.

Survival	Factor	Hazard Ratio (95% C.I.)	*p* Value
CNS9204 Event-free survival (EFS)	BLBP expression ^a^	2.08 (1.01–4.26)	**0.04**
	Location (PF vs. ST)	7.09 (0.94–53.32)	0.06
	Resection (Incomplete vs. Complete)	1.03 (0.51–2.07)	0.94
	WHO Grade (Grade III vs. Grade II)	0.81 (0.39–1.66)	0.55
CNS9204 Overall Survival (OS)	BLBP expression ^a^	2.81 (1.2–6.87)	**0.01**
	Location (PF vs. ST)	0.00 (0.00 → 10e10 1 × 10^11^)	0.97
	Resection (Incomplete vs. Complete	1.05 (0.47–2.34)	0.91
	WHO Grade (Grade III vs. Grade II)	1.35 (0.58–3.13)	0.97
CNS9904 Overall Survival (OS)	BLBP expression ^a^	5.37 (1.70–16.94)	**0.004**
	Location (PF vs. ST)	1.84 (0.52–6.47)	0.34
	Resection (Incomplete vs. Complete)	1.08 (0.35–3.31)	0.88
	WHO Grade (Grade III vs. Grade II)	2.32 (0.51–3.55)	0.14

^a^ positive versus negative. Bold: emphasise the significant values.

## Data Availability

All data is contained within the article or Appendix A.

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
