# Peer review of "BLBP Is Both a Marker for Poor Prognosis and a Potential Therapeutic Target in Paediatric Ependymoma"

_cancers, 2021, doi:10.3390/cancers13092100_

Round 1
Reviewer 1 Report
I thank the authors for taking into account most of my sugesstions and I feel that they significantly improved the quality of manuscript.
Reviewer 2 Report
The authors have considered all my concerns and the revised version of the manuscript is improved substantially. The manuscript is a nice piece of work and should be considered for publication.
This manuscript is a resubmission of an earlier submission. The following is a list of the peer review reports and author responses from that submission.
Round 1
Reviewer 1 Report
The authors present a very interesting manuscript and provide evidence for BLBP being new prognostic marker and a potential new therapeutic target in Ependymoma, one of the deadliest forms of pediatric cancer. However, at some points the study seems preliminary and few issues should be addressed before publication.
1.-Despite the relevance that the distinct molecular subgroups of EPN may have for the understanding of the disease and addressing new forms of therapeutic intervention, this aspect is poorly addressed through the manuscript. Thus, I'd suggest to take into account this matter in the following sections:
a) Methods: Please define the molecular subgroupg of the cell lines selected for the study.
b) Results: Include in Figure 1 data that is mentioned in the discussion about the expression levels of BLBP in the tissues analyzed. Data mining could also be explored in gene expression databases to visualize the patter of mRNA expression in the different EPN subgroups.
2.-For clarity of the results, please highlight those variables with a significant p value.
3.-Regarding the qPCR expression values in the cell lines analyzed, do the authors presume that BLBP is expressed only in a faraction of cells or a more homogeneous distribution? Then, if this is the case, are those cell lines enrichsed in "stem cells"?
4.-To elucidate the functional role of BLBP in paediatric ependymomas, the most obivous experiment would be to perform loss of function experiments with siRNA/shRNA or CRISPR. The indirect pharmacological approach it is a very valid therapeutic strategy but, owing to the myriad of PPAR-gamma regulated genes, nto the best one to elucidate the BLBP function. Thus, to claim that BLBP would be a therapeutic target, the effects of targeting directly BLBP must be addressed.
5.-The authors use three different PPAR-gamma antagonists, but they impact differently on EPN cells. Please discuss.
6.-In Figure 3a, please double check the nomenclature of the Y axis. The 2elevated at minus deltadeltaCt usually is refered to a "control" and the results are given in a fold change. Here it seems that the expression value is only referred to the housekeeping gene. Please, clarify.
7.-In figure 3, why "representative images of the treated cells" are only shown for DKFZ-EPN1 cells? if they are cultured in neurosphere medium, shound't be they called DKFZ-EPN1NS? Please clarify. Since different methods are used to calculate viability in the same figure, it would be helpful to have it indicated in the figure legend.
8.-Figure 4 could be better structured. Perhaps in this figure, the main message should be the effects of PPAR antagonists on migration and invasion of EPN cells. Please also show here the representative images for both cell types for consistency. The results of sensitization of other chemotherapies, albeit very interesting, could be in a different figure.
Furthermore, the representation of the wound closure is not very intuitive. Higher T1/2 represents less migration? Perhaps representing the % of wound closure (in area) would be easier to interpret, and also give an idea, if migration is delayed or blocked.
9.-For Figure 4 and 5, I'd suggest a re-structuration. All experiments of migration/invasion could be in the same figure and those related to chemosensitization in another one.
10.-One very important missing piece of information is regarding the feasibility of using this therapeutic approach in vivo. Inducible knockout of BLBP in orthotopic xenografts will give a definitive proof that BLBP is an interesting therapeutic target in EPN. Furthermore, the potential use of PPAR antagonists or DHA for brain tumors needs to be discussed.
11.-Potential downstream effectors of BLBP are only mentioned in the discussion. Could any of these regulated genes/pathways we assessed in the in vitro models? This would help to decipher whether the observed phenoytipic effects are BLBP dependent and not due to targeting any of the multiple PPAR-regulated genes.
Reviewer 2 Report
In this manuscript, the authors showed that BLBP is an independent predictor of poor survival in pediatric ependymoma, and treatment with PPAR antagonists or DHA may represent effective novel therapies, preventing chemotherapy resistance and invasion in pediatric ependymoma patients. My specific comments are as below.
1. The authors concluded that aberrant expression of BLBP could modify cell migration, proliferation, and chemo-resistance in the pediatric ependymoma cell line. However, their results do not directly prove that BLBP is a key factor controlling the nature of cancer. To clarify that BLBP controls cell proliferation, migration ability, and infiltration ability of the ependymoma cell line, it is better to conduct a knockdown experiment using siRNA (a knockout experiment using CRISPR-Cas9 should be better) in a cell line with a high BLBP expression level (DKFZ-EP1 and EPN7, etc.) and/or an overexpression experiment in a cell line with a low BLBP expression level (BXD1425-EPN and EPN1).
2. The authors also concluded that inhibition of BLBP expression via the PPAR-g antagonist GW9662 and the DHA could suppress cell migration, proliferation, and chemo-resistance in the pediatric ependymoma cell line. However, their experimental results do not appear to directly prove that the effects of these antagonists are mediated only by suppression of BLBP. BLBP should be introduced into cell lines treated with GW9662 or DHA to examine whether the effects of these antagonists are canceled.
Minor comment:
1. The expression level of BLBP protein should be examined by Western blotting in Figures 2b and 2c.